# Risks and Barriers in Substitute Care for the Children of Parents with Serious Mental Illness: A Mixed-Method Study in Kerala, India

**DOI:** 10.3390/healthcare10122408

**Published:** 2022-11-30

**Authors:** Saju Madavanakadu Devassy, Lorane Scaria, Lynette Joubert

**Affiliations:** 1Department of Social Work, Rajagiri College of Social Sciences (Autonomous), Kochi 683104, Kerala, India; 2Rajagiri International Centre for Consortium Research in Social Care, Kerala India, Rajagiri College of Social Sciences (Autonomous), Kochi 683104, Kerala, India; 3Honorary Principal Fellow, Department of Social Work, Melbourne School of Health Sciences at the University of Melbourne, Parkville 3010, Australia; 4Department of Social Work, Melbourne School of Health Sciences at the University of Melbourne, Parkville 3010, Australia

**Keywords:** parental mental illness, children, risk factors, family care, India

## Abstract

Background: Mental illness in parents impairs their parenting capability, which has a lifelong detrimental impact on their children’s physical and psychological health. In the current Indian context, due to weak social security nets, family is the only plausible intervention to ensure adequate substitute child care. Therefore, this study explores various risk factors and barriers to providing substitute family care. Methods: We used a mixed-method approach to gather information from 94 substitute family caregivers. Quantitative screening data were collected from four hospitals using a clinical data mining tool and an interview guide to gather caregiver perspectives on economic, familial, and social risks and barriers associated with caring. We used thematic analysis to consolidate the qualitative findings. Results: Most of the substitute caregivers were females from low-income households. The study identified 11 sub-themes and 23 specific themes associated with risks and barriers to substitute care. These themes fell into four broad areas: economic, familial, school-related risks, and specific cultural and service access barriers. Focus on economic interventions is likely to result in strengthening the substitute family caregiver. Conclusion: The paradigmatic shift of treatment focus from the patient to the entire household would benefit the children just as it does the patient.

## 1. Introduction

Globally, 68% of women and 57% of men with Serious Mental Disorders (SMD) are parents [1], and 23–32% of them are primary caregivers and are frequently admitted to mental health treatment facilities [2,3]. In India, one in seven people is affected by one or more mental illnesses, and the disease burden has almost doubled since 1990 [4]. The mental health of their children is at increased risk due to several reasons, including genetic predispositions associated with parental illness [5], direct exposure to parental psychiatric symptoms, and the influence of mediating factors, such as a damaged marital relation or inter-partner violence [6] and a dysfunctional family environment [7]. Disrupted life routines, frequent hospitalization of parents, and social stigma are some of the determinants of a dysfunctional family environment [8]. Co-occurring psychosocial adversities, such as poverty, downward social and economic mobility, stigma related to illness, and strained parent–child relationships are other major areas of dysfunction specific to Indian families [9].

Family dysfunction and experiences of greater material deprivation limit their children’s developmental opportunities and increase the risk of child maltreatment and neglect [10]. These familial and economic challenges were found to be the predictor of academic underachievement [11,12], behavioural and developmental issues, as well as physical and psychological ill-health [13], lesser competency than their peers [14], and lower levels of health-related quality of life among children [15]. This affects emotional health, interpersonal engagement, family functioning, and parent–child interaction [16]. These issues extend into adulthood, causing social or occupational dysfunction, an increase in psychiatric morbidity, and drug or alcohol abuse [16]. However, researchers and mental health professionals are often insensitive to the needs of these vulnerable children and prefer to concentrate solely on the patients.

Only one-third of the mentally ill parents in India raise their children, and the rest are cared for by their spouses, grandparents, or other family members [8,17]. Although the needs and issues of the caregiver might vary based on the structure of the family and also the child’s relationship with them [9], family support interventions are crucial for ensuring better care for children. Furthermore, the social changes brought on by changing family norms driven by epidemiological and demographic transitions are continuously challenging the altruistic mindset of substitute family caregivers [18]. Family caregivers would be encouraged to continue caring for these children if child-caring needs are identified and actively supported [19].

The current study aims to explore various risk factors and barriers in providing high-quality substitute care for the children of seriously mentally ill parents. The findings can potentially bring paradigmatic changes in the mental health treatment provided in India by shifting the focus from the medical to the psychosocial model, and the patient to the household. The current study also aims to generate knowledge on the familial aspects of parental mental illness and how families navigate parenting challenges in Indian contexts to inform policies, programs, and strategies where the Indian data is sparse [9].

## 2. Materials and Methods

### 2.1. Design

The mixed-method study was conducted in four purposively selected psychiatric inpatient care facilities situated in three districts, representing the socioeconomic and cultural diversity of the people of Kerala: Ernakulum (urban), Idukki (rural), and Thrissur (semi-urban). Information was obtained from the substitute caregivers of children with a parent with mental illness. The face-to-face interviews were conducted from December 2019 to February 2020 at the residences of eligible participants. To maintain confidentiality, each respondent received a unique ID.

### 2.2. Participant Recruitment and Criteria

Patients and their caregivers were identified from hospital medical records, and substitute caregivers (taking care of the children) were contacted over the phone to obtain an appointment. Selection criteria for inclusion were patients who were hospitalised for severe mental illness, having at least one child between the ages of 0 and 18 at the time of diagnosis, and having a spouse or a close relative available for the care of the patient’s children in their absence. We extracted patient demographic data from the hospital medical records using a clinical data mining tool, and a list of eligible participants was created. Family bystanders of the patients provided the address of the substitute caregiver. Out of 221 eligible participants, 94 (42.54%) consented to be part of the study. Participants were recruited from the community at their homes based on the information from the bystanders and after obtaining informed consent. See Figure 1 for more details on recruitment.

### 2.3. Sample

Caregivers in the current study were either a mother (*n* = 22), father (*n* = 24), grandparent (*n* = 25), or other relatives (*n* = 23). Seventy-six per cent of the caregivers were females. Clinical data mining generated the profiling of the parent and caregiver. The study included the caregivers of children of 40 male and 54 female diagnosed parents, and the most common mental disorders reported were bipolar affective disorder and schizophrenia (63.4%). The mean age of children was 13.9 (Table 1).

### 2.4. Supervision and Training

Six second-year psychiatric social work students, divided into three teams of two each, conducted the interview. The pairing of the teams was conducted particularly to ensure inter-rater reliability. Standard operating procedure manual-based training was provided to them prior to the interview. The primary screening tools and interview guides, as well as field operations, were covered in the standard operating procedure manual that guided the training. Because the research was not intended to be an intervention, the trainees received training on the purpose and objectives of the research in detail. The student researchers used role plays and mock interviews to practice conducting interviews and to ask culturally sensitive questions. They contextualised the tools to make them engaging and culturally appropriate. Each interview lasted for 40–50 min.

### 2.5. Data Mining Tool

The key aspects covered in the data mining tool were demographic information, parents’ type of disorder, medical history, and the availability of substitute parenting. Demographic variables consisted of age, gender, relationship with the child, age of the caregiver, education, occupation, and marital status of the caregiver. Regarding the parent with mental illness, the gender, the type of illness, and marital status were recorded. The tool also contains information regarding the selection criteria used during the participant recruitment specified in earlier sections.

### 2.6. In-Depth Interview Guide

The interview guide was used to collect information regarding the economic, familial, cultural, community, and social context of caring for the child. The economic domain probed the socio-economic profile of the families, the financial impact of illness on the children and the family, and the burden of care due to financial hardships. Familial aspects were explored through probes on interpersonal relationships, attachment to the caregiver and social relationships, family caregiver burden, and the multidimensional impact of illness on children. The family and community beliefs, availability of formal and informal support networks, and stigma and discrimination experienced were explored as the social aspects of mental disorders. Enquiries were also made about relationships between siblings, members of the extended family, the child’s social circle, and relationships with schoolmates.

### 2.7. Translation of Data Collection Tools

The principal researcher and two other researchers double-translated the interview guide into Malayalam (the local language), and then an independent researcher back-translated it into English. Items were carefully evaluated to detect any potential language or meaning barriers within any specific ethnic population at each study site to avoid any miscommunication during the interviews.

### 2.8. Data analysis

#### 2.8.1. Quantitative

Quantitative data analysis was performed using IBM SPSS version 25. Descriptive statistics were performed to analyse the frequencies and percentages of the demographic information obtained through clinical data mining.

#### 2.8.2. Qualitative

*Transcribing the data*: All six trainees transcribed the qualitative data, indicating all non-verbal information in the transcript. In the presence of the principal investigator, the research trainees examined the Malayalam content. They also ensured that the intensity of the experiences was not lost when translating the qualitative content from Malayalam to English. Verbatim quotes, informal remarks and observations, and interpretations of participants’ comments were included in the handwritten notes. These notes were added to the transcripts to provide context and improve the researcher’s comprehension of the transcript. After agreeing upon the translation, the data were entered into the Excel file. The typed-out data was stored on an external hard drive on the cloud storage system used by the college to ensure data security.

*Coding the Data*: The research students categorised and sorted the data for analysis once it was collected, transcribed, and translated. Themes were generated inductively by annotating them with simple codes representing their experiences based on the key phrases. A few initial categories were refined later. The study team scanned the transcripts as coded to examine what patterns emerged from the data and to note further thoughts or ideas. The researchers then assigned colours to the codes. Data, coded and sorted by theme and domain, were piled up to remove irrelevant or unnecessary items. The research team used deductive reasoning to create an initial coding system, and they then went through the procedure again to improve, expand, or discard some of the initial categories. The initial analysis was done in the semantic level to understand the strength of these observations. The prevalence of each sub-theme was explored to prioritise the caregivers’ needs. The quotations were reviewed by the research team, linguists, and subject experts to ensure they retained their original meaning. The authors carefully considered whether the codes were appropriate for each item before consenting to them. Emerging patterns were grouped to produce meaningful themes and mapped to generate the main themes and subthemes.

*Thematic analysis*: We used realist and interpretative phenomenological analysis based on Braun and Clarke’s thematic analysis guidelines [20] to analyse the qualitative data. We used critical realism, which recognises both objective observations and subjective perceptions. We attempted to understand the experiences within the constraints of the social, cultural, psychological, and economic contexts of the respondent. This helped to consolidate the caregivers’ experiences of children whose parents had a severe mental illness. The themes centred around economic risks, with subthemes on the economic drift, impoverishment due to treatment expenditure, and malnourishment. The social aspect, especially the family risks, was related to disturbed family patterns and marital adjustments, poor parent–child interaction, and caregivers’ distress due to their multiple roles and responsibilities. The school domain explored the details of the insensitive school system and exclusion and exposure to negative coping behaviours. The challenges in navigating resources for these children centred around culturally driven social exclusion and access to statutory and informal support nets.

### 2.9. Ethical Approval

This study received Institutional Ethics Committee approval from Rajagiri College of Social Sciences with Reference No: RCSS/IEC/007/2019. Before the interview, written informed consent was obtained from each participant and the hospitals for data mining.

## 3. Results

Table 2 describes the various challenges faced by caregivers in taking care of the children. Major socio-economic risk factors were low income, reduced employment opportunities for diagnosed parents, financial hardships, and associated disturbances in family systems. Familial risk factors were impaired parent–child interaction, and social and cultural risk factors were limited access and awareness about support systems and services, lack of adequate support and integration at schools, and reduced neighbourhood cohesion owing to rampant stigma and discrimination associated with mental disorders.

Major themes derived from the qualitative thematic analysis are included below.

### 3.1. Theme 1. Economic Risk Factors for the Care of Children

#### 3.1.1. Economic Drift and Its Effect on Substitute Caring

The majority of the caregivers belonged to low-income households without adequate housing or other basic amenities. The majority of households earned between Rs. 3000–6000 ($40–$80) per month, with the exception of two that made Rs. 200,000 ($2700) each month. Mental illness hindered the parental ability to find and sustain employment, which disturbed their financial stability, leading to impoverishment. The impoverishment resulted in extreme economic hardship, absolute poverty, poor housing conditions, marginal social status, and cultural alienation. Most caregivers experienced difficulty meeting the family expenses, patient’s treatment expenses, and child’s educational and health care needs. An old woman caregiver (aged 60, farming cardamom on a small piece of land) responded: *“I am not capable of providing better health care to my daughter or meeting the educational needs of my grandchild. I cannot afford to buy new dresses for him”*.

#### 3.1.2. Impoverishment due to Treatment Expenditure

Financial resource shortage was a barrier to medication and treatment adherence, though treatment adherence is a crucial component of home-based patient management and has implications for recovery, relapse prevention, and community-based rehabilitation. One of the caregivers mentioned, “We spend between Rs. 2500 to Rs. 3000 ($40–$50, which is half of the family’s monthly income) a month on medicines, and our income is inadequate to meet the sustenance needs of a five-member family. So, we stop the treatment when my husband gets better. But he frequently relapses, which pushes up the treatment costs which further deprives the family”. (38-year-old woman working as a housemaid).

#### 3.1.3. Malnourishment in Children

Families with a father who has SMD and is the main breadwinner experience sporadic unemployment, especially during the symptomatic stage of the illness, putting the family in a precarious financial situation. Lack of income leads to malnourishment, consequently preventing the children from reaching their full developmental potential:

*“I have to take care of my ill husband and his medical expense. I am always concerned about the inability to provide adequate nutritious food and meet educational expenses for my children”*. (32-year-old woman with two children).

### 3.2. Theme 2. Social Domain- Familial Risk Factors

In households, the caregiver’s responsibility is shared between the spouse, grandparents, and other extended family members. The maternal grandparents and extended family care for the children of mothers with SMD. However, grandparents frequently felt incompetent to nurture their grandchildren: *“I am old, I am unable to work and support my grandson, who is always upset by his mother’s condition. He is highly agitated and restless. I don’t know how to handle him”* (64-year-old woman, farming in a small piece of land).

#### 3.2.1. Disturbances in Family Systems and Marital Adjustments

Parental mental disorders cause significant disruptions in a family’s ability to adjust, leading to issues in childcare. A father caring for their child felt anxious about their future (32 years old, daily wage worker) said, *“I am unable to create a conducive peaceful atmosphere at home for her, even though she is good in her studies”*.

However, some of these children felt the family environment was too stressful to tolerate: *“The only time I feel relaxed and happy is the time with my friends. I don’t feel like coming back home”* (14-year-old boy with a father suffering from schizophrenia told his mother, who complained about his late return home.). Another boy, aged 13, said to his father, *“I would be happier if there were classes on Saturdays and Sundays…. I don’t want to be at home with that nagging mother”*.

Notably, with children between the ages of 4 and 8 years old, several families kept them away from their ill parents to hide the fact of their parental mental illness. The children often experienced neglect and inconsistent care at home and felt deprived of a sensitive, responsive, and secure environment: *“Sometimes my husband is too angry and hostile towards him (child aged 16), making him sad and thereby disrupts the relation”* (41-year-old mother, a daily wage worker). Many caregivers expressed the view that the parent’s hospitalization is frequently a danger to the stability of the family and their access to financial and social resources, especially if the children must move to another home, such as a relative’s home, throughout the course of treatment, especially for single-parent households.

#### 3.2.2. Poor Parent–Child Interaction

A few children tended to avoid parental interaction due to inadequate knowledge about parental mental disorders: *“My father does not respect others and frequently gets into unnecessary arguments. I don’t feel comfortable in public when he is with me. I am embarrassed to go out with him because people make fun of us”* (Commented by a 12-year-old girl to her mother.).

The child’s interaction with the parent who has SMD is reduced after knowing about the illness, partly due to the child’s fear of being attacked by the sick parent or because of stigma and discriminatory social conventions:

*I’m afraid to go near papa, and I don’t feel comfortable expressing my needs to him because he is very rude and angry at me. I think he doesn’t like me.* (13-year-old boy told his mother). Another respondent with a boy child of 8 years stated that *“My child is terrified of his father due to the violent behaviour and hostile attitude. He (Child) had dreams in which his father attacked and tried to kill him”*.

#### 3.2.3. Caregivers’ Distress due to Multiple Roles and Responsibilities

Fathers who have SMD often receive more attention from both families, in-laws, and their own families. In contrast, support from the spouse’s family is uncommon for mothers who have SMD: *“I’m worried about the way my children are being treated by my in-laws. Who will take care of them after my death? I fear that they (in-laws) may reject and will not give my rightful property shares or entitlements to my children”*. (32-year-old woman patient with uninvolved husband, having two children aged below 8 years).

A grandmother (61-year-old widow, farming in a small plot of land) said, *“He (son-in-law) left her and their two children with me. He or his family is bothered about them. I am not sure how far I can care for them”*.

Several mothers felt bad for not being able to give their children *“adequate care and support”*. *“I am not satisfied with the amount of time spent with my children… I am a failure as a mother”* (29-year-old woman with two children aged below 5 years). In situations where the mother is mentally ill, maternal grandparents, and relatives from the maternal side care for the children.

### 3.3. Theme 3. Unsupportive School System

#### 3.3.1. Insensitive School System and Exclusion

A few children were reluctant to go to school due to the discrimination and stigma they experienced at school. They often disengage, and never disclose their parental mental illness to others out of fear of being stereotyped.

One of the respondents shared, *“my daughter feels stigmatized by her father’s condition… she never goes in public… never gets along with friends… never attends any family functions and always sits ruminating”*.

A few children even stopped going to school because of the impact of parental illness on their lives. The mother of an eight-year-old boy said, *“My child discontinued the school because his classmates nicknamed him as “mad man’s son” … he hates going to school”*.

Children of parents with mental illness tend to feel excluded at school. A 7-year-old boy child often told his mother, *“My father is “mad”... other children do not include me in their groups… I don’t want to play with them (his friends)”*.

Children’s involvement in activities outside the home provides them respite from the stressful family environment, but girl children are deprived of that: *“The boys are less socially withdrawn than girls because they are involved in games and other activities”*. (The mother of a 14-year-old female child observed).

#### 3.3.2. Exposure to Negative Coping Behaviours

Parental mental disorder is also associated with risky behaviour among a small group of adolescent boys. They chose to socialise with adults who were older than they were, where they got exposed to risky behaviours such as drug use, alcohol, truancy, and gang activity: *“He doesn’t care what is right or wrong… his gang is dangerous; they introduced marijuana… now he is peddling it. He won’t listen to us, no matter how hard we attempt to correct him”*. (Reported by a maternal grandmother caring for a 16-year-old grandson. His father left them after his mother was diagnosed with schizophrenia).

### 3.4. Theme 4. Challenges in Navigating Parenting Resources

#### 3.4.1. Culturally Driven Social Exclusion

A few respondents talked about social isolation from the members of the dominant communities: “We are constantly teased and excluded from the community because we live in an area where………. (a major religion) dominates. They even discourage their children from playing with our children” (28-year-old wife of a person with schizophrenia). A few of the socially excluded respondents reported, “after the diagnosis (of mental illness), the neighbours stopped coming over” (28-year-old wife of a patient with schizophrenia), and “seldom ever spoke to us” (49-year-old mother of a patient with the bipolar affective disorder), and “never invited us to any gatherings” (32-year-old wife of a patient with schizophrenia). Another added that “earlier they brought special dishes for festivals and special occasions, but they stopped everything… they refused to accept ours either” (34-year-old wife of a patient with bipolar affective disorder). As a result, the children felt isolated, unsupported, and deprived of developmental opportunities.

#### 3.4.2. Inadequate Statutory and Informal Support Nets

Unfortunately, the majority of respondents were uninformed of the statutory and informal services. One of the caregivers responded, *“I don’t have a full-time job... I have two cows, and that is my only income. I had to borrow money to meet the expense of the child’s education. I don’t get any support from the panchayat (local self-government), relatives, or others”*.

However, some receive assistance from informal support organizations, including Non-Governmental Organizations (NGOs), churches, neighbourhood self-help groups, teachers, neighbours, extended families, and a few clubs. However, these services are only offered to a smaller number of families, and the little that they do receive is insufficient

## 4. Discussion

Our study explored various risks and barriers faced by substitute caregivers of children of parents with mental disorder diagnosis and treatment. In this scenario, the child’s care is frequently assumed by the spouse or other family members, such as grandparents or relatives, who are mostly females. The study presented the challenges in child care under four major themes and several sub-themes. Among them, the family and the school were crucial environments for ensuring children get opportunities to flourish, and problems in either of these had a negative impact on the children. However, economic stability and cultural inclusion are the prerequisites for modifying the family and school environment favourably.

We noticed a recurring pattern in families where the children of mothers with a diagnosis were cared for by the maternal grandparents and other extended family members on the maternal side. These substitute caregivers, the grandparents in particular, often felt incompetent in providing their grandchildren with adequate nurturing. Additionally, one of the reasons for the child’s poor positive interaction with the diagnosed parent is their mental health illiteracy. Our study indicated that children prefer to engage with their parents pleasantly and positively; however, it always distresses them when the parent becomes hostile and irritated for no apparent reason. This finding is consistent with other studies [21]. Several studies have described the undeniable association between mental illness-related social exclusion and family deprivations [21,22]. The symptomatic behaviour of the diagnosed parent often strains the relationship with others, such as the extended family, neighbours, and the community at large. Additionally, the mental health illiteracy of the community, neighbours, and others adds to the distress by cutting off contact with these families, presuming that the entire family is suffering from mental illness and, as a result, generalising the patient’s abusive, suspicious, and erratic behaviour to the entire household.

We included the school barriers to show that if the school is supportive, it would be protective against caregiver stress, especially the worry regarding the child’s future, since academic success is a requirement for better jobs, higher salaries, better living conditions, and, therefore, economic well-being. Rampant stigma and discrimination, and mental health illiteracy are barriers to the integration of children into the school system, which is otherwise a protective factor for their mental health. Children experience peer isolation and loss of developmental opportunities. As a result, some children exhibit school refusal, and others even turn to risky behaviours, such as drug use, alcohol abuse, and truancy. The study also found that children who are socially isolated and reclusive at school are more likely than integrated children to experience emotional and behavioural issues. The latter get more acceptance, better opportunities, and support, whereas the former alienate themselves and feel isolated [23]. Therefore, a supportive school system would be a valuable resource for the substitute caregiver.

The sensitivity and willingness of schools to deliberately integrate them into the school system can serve as a buffer to shield these children from the detrimental effects of unfavourable familial and societal environments. India is one example of a collective society with strong family and community bonds. They experience social cohesion in a supportive neighbourhood, which promotes trust, safety, and community inclusion. Social inclusion is essential for lowering familial stress levels, facilitating better access to resources for addressing material deprivations, and ensuring a sense of support. Thus, this study motivates mental health professionals to consider the school as a cost-effective platform to deliver lessons on mental health literacy and social inclusion.

The economic drift caused by parental mental illness places a significant financial burden on caregivers and compromises their ability to provide for the needs of the children. Our study reinforces the relationship between poverty and mental health and its effects on children [24]. Additionally, poverty and huge out-of-pocket expenditure are barriers to treatment adherence. This is critical in the context of research evidence showing that parental treatment adherence and medical compliance are vital to reducing distressing behaviours, psychiatric symptoms, and hospitalisation-related parental absenteeism [25]. Parental mental illness-induced impoverishment and unsupportive family environments hinder the growth potential of children as the home plays a crucial role in providing a positive trajectory of mental capital across the lifespan [26]. Additionally, the most important barrier for caregivers in caring for these children is material deprivation. Therefore, the provisions for universally available, easily accessible formal and informal support and specific services would ultimately benefit the children to optimise their developmental potential to become contributing citizens of the country.

The study contributes to the growing body of Indian literature on contextualizing the multisystemic risk factors associated with child care. In the absence of formal foster care in India, most of the children of parents with mental illness are cared for by substitute caregivers. As a result, this study assists mental health teams to prioritise the substitute caregiver’s specific needs, which will aid in developing tailored interventions. The study recommends prospective agencies that could assist substitute caregivers in caring for these children, such as schools, and formal and informal organisations. It also highlights the inadequacies of the present policies, systems, and services in India that are designed to safeguard child and family mental health. Most importantly, the study highlights the need for enhanced, culturally appropriate mental health awareness campaigns to counter the stigma and discrimination of mental illness in society. The study had some limitations, as well. The recruitment of the participants from hospital-based settings might affect the generalisability of the findings in wider community settings. The primary caregiver responses might have been skewed because of the possibility of the dominance of negative experiences over positive ones.

## 5. Conclusions

Parental mental disorders inflict a severe toll on children throughout their lives. Immediate or extended family members can feel burdened and stressed when they are left with the multiple responsibilities of looking after the mentally ill person and their children. To better equip the substitute caregivers to take care of this vulnerable group, economic support and school should take precedence over family-focused interventions. Study findings demonstrated that household-based mental health treatment would benefit the children most, just as patients benefit from family-focused and community-based interventions. These insights have the potential to guide future practice, policy, and research to ensure favourable growth environments for children.

## Figures and Tables

**Figure 1 healthcare-10-02408-f001:**
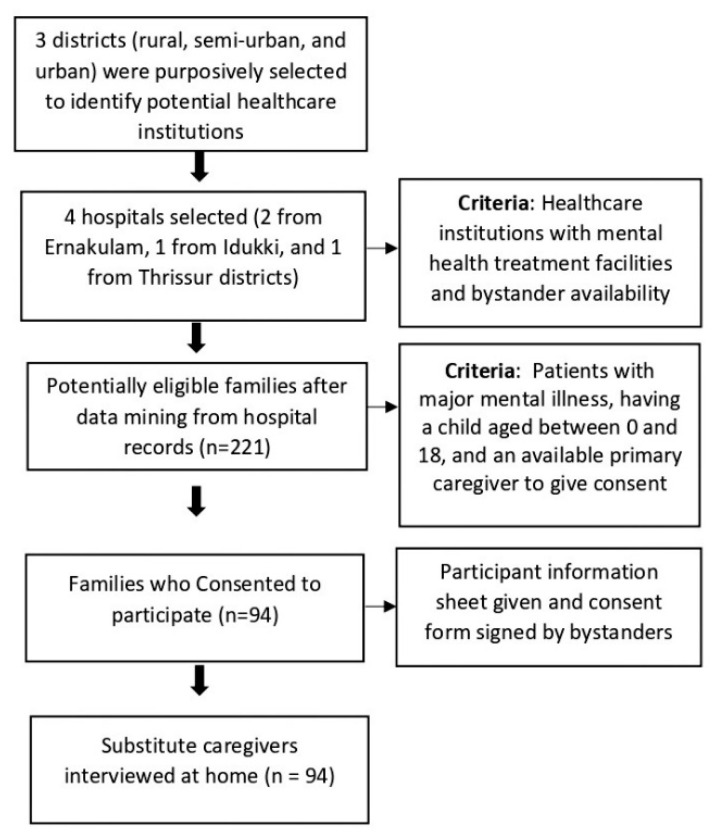
Participant recruitment for the study.

**Table 1 healthcare-10-02408-t001:** Demographic profile of the parent with mental illness and their children.

Demographic Profile	Frequency	Percentage
**Profile of the caregiver**		
*Gender*		
Male	22	23.4
Female	72	76.6
*Relationship with the child*		
Mother	22	23.40
Father	24	25.53
Grandparents	25	26.60
Others	23	24.47
*Age of the Caregiver*		
Below 40	54	57.4
Above 40	40	42.5
*Employment*		
Employed—full-time/part-time	34	36.2
Unemployed	60	63.8
**Profile of the parent with mental illness**		
*Gender*		
Male	40	42.5
Female	54	57.5
*Diagnosis*		
Bipolar affective disorder	32	31.7
Depression	18	17.8
Personality disorder	14	15.8
Schizophrenia	30	31.7
*Marital status of the caregiver*		
Married	81	86.2
Separated/Divorced/Widowed	13	13.8
**Profile of the child**		
Mean age of the children	13. 9 (5.7)	
Gender of the child		
Male	43	45.7
Female	51	54.3

**Table 2 healthcare-10-02408-t002:** Risk factors in taking care of children as reported by the caregivers.

Major Themes	Sub-Themes	Specific Risk Factors	Frequencies (%)
Economic risk factors	Economic drift	Low-income households	85 (90.4%)
		Reduced employment opportunities for parents due to their mental illness	62 (65.9%)
		Low educational attainment of parents and economic drift	59 (62.8%)
		Inadequate housing	61 (64.5%)
	Economic deprivation	Financial hardships	84 (89.4%)
		Families living in absolute poverty due to parental mental disorders	9 (9.5%)
	Malnutrition	Nutritional deficiency/malnourishment in children due to lack of money	20 (21.2%)
	Treatment non-adherence	Failure to adhere to treatment due to financial difficulties	16 (17%)
Familial risk factors	Disturbance in family systems and marital issues	Financial difficulty-induced disturbances in family systems and marital adjustments	65 (69.1%)
		Marital conflicts and poor interpersonal relationships because of mental illness	7 (7.4%)
	Poor parent–child interaction	Impaired parent–child interaction due to unavailability, parental aggression, and hostility	26 (27.7%)
		Lack of emotional attachment with the parent who is mentally ill	25 (26.6%)
		Parental aggression and hostility prevent children from developing warm and positive interactions with their parents	15 (15.9%)
	Caregiver burden	The feeling of unbearable burden and stress associated with caregiving in the substitute caregiver	24 (25.5%)
		The sole responsibility for the care of children leading to burnout and stress	24 (25.5%)
Inadequate support from School system	Insensitive school system	School refusal due to fear of isolation	10 (10.6%)
		The disinterest of children in attending school, fearing exclusion and discrimination	7 (7.4%)
		Dropping out of school due to parental mental illness	2 (2.1%)
	Lack of inclusive school system	Reduced academic performance, social relationships, and peer interactions due to parental mental disorders	29 (30.8%)
Challenges in navigating parenting resources	Social exclusion	The reluctance of neighbours to allow their children to play with these children	8 (8.5%)
		Reduced neighbourhood interaction due to parental mental illness	21 (22.3%)
		Rampant stigma and discrimination and unsupportive socio-cultural environment in rural areas	26 (27.6%)
	Issues in accessing services	Inadequate awareness and access to formal and informal support systems and services	77 (82%)

## Data Availability

The data that support the findings of this study are available from the corresponding author, S.M.D., upon reasonable request.

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
