# Peer review of "Risks and Barriers in Substitute Care for the Children of Parents with Serious Mental Illness: A Mixed-Method Study in Kerala, India"

_healthcare, 2022, doi:10.3390/healthcare10122408_

Round 1
Reviewer 1 Report
This study used a mixed-method approach and explored various risk factors and barriers to providing substitute family care from 94 substitute family carers. The authors identified 11 sub-themes and 23 specific themes associated with risks and barriers to substitute care. This is a timely and interesting topic to shift treatment focus to the entire household among Indian patients. Below are some comments to improve the manuscript.
Introduction:
1. The authors reviewed studies on the prevalence of serious mental health disorders and some of the determinants of a dysfunctional family environment. What are the prevalence and factors in India?
2. What is the rationale to suggest shifting the focus from the patient to the household?
3. What is your research question and hypothesis?
4. The introduction section should be more comprehensive in general.
Methods and results:
5. What research tool did you use to analyze the qualitative data?
6. The results section is generally well-written.
Discussion:
7. The authors promote a household focus for treatment. What is the role of the school in this shift?
8. The authors are suggested to relate your results to the Indian context to discuss your research implications.
9. Please add a brief discussion of the limitations of the current study.
Author Response
This study used a mixed-method approach and explored various risk factors and barriers to providing substitute family care from 94 substitute family carers. The authors identified 11 sub-themes and 23 specific themes associated with risks and barriers to substitute care. This is a timely and interesting topic to shift treatment focus to the entire household among Indian patients. Below are some comments to improve the manuscript.
Author response: Thank you for agreeing to review our manuscript. Your valuable suggestions are greatly appreciated, and we have modified the manuscript based on your comments.
Introduction:
- The authors reviewed studies on the prevalence of serious mental health disorders and some of the determinants of a dysfunctional family environment. What are the prevalence and factors in India?
Author response: We have updated the introduction section with the prevalence and risk factors of mental illness in India.
- What is the rationale for suggesting shifting the focus from the patient to the household?
Author response: We have further discussed this in the discussion section.
- What is your research question and hypothesis?
Author response: The aim of the study is added at the end of the introduction.
- The introduction section should be more comprehensive in general.
Author response: We have modified the introduction section to make it more comprehensive, updating newer references and structuring the content.
Methods and results:
- What research tool did you use to analyze the qualitative data?
Author response: Content thematic analysis was employed to qualitatively analyze the various risk factors of substitute care for children. Analysis was based on Braun and Clarke’s guidelines for analysis. Further we have also added a few sentences about the philosophical considerations.
- The results section is generally well-written.
Author response: Thank you.
Discussion:
- The authors promote a household focus for treatment. What is the role of the school in this shift?
Author response: We included the school barriers to demonstrate that if the school is supportive, it would be protective against family stress, especially the worry regarding the child’s future, as academic achievements are prerequisites for better jobs, higher income, better living conditions and, thereby, economic well-being. Therefore, a supportive school system would be a great resource for the substitute caregiver.
We have discussed this in discussion.
- The authors are suggested to relate your results to the Indian context to discuss your research implications.
Author responses: We have included a paragraph on research implications towards the end of discussion
- Please add a brief discussion of the limitations of the current study.
Author responses: Limitations have been added.
Reviewer 2 Report
This paper is of interest in exploring the risk factors and barriers to providing substitute family care for children whose parents have mental illness. To conduct this study, the authors use a descriptive quantitative and qualitative research design with an interesting sample of 94 participants (2.54% of the eligible participants), all of whom are substitute family caregivers in India. The authors collect caregivers' perceptions of the economic, family and social risks and barriers associated with caregiving. The majority of the substitute caregivers were women from low-income households. They suggest the identification of 11 specific sub-themes associated with the risks and barriers of substitute care. These sub-themes are grouped into four broad themes: economic, family, school-related risks, and cultural and service access barriers. It concludes with the need for a paradigm shift of the treatment approach from the patient to the whole household to benefit children and patients.
On the other hand, the literature review is interesting, and both primary and secondary scientific sources are collected. However, although the paper succeeds in addressing an acceptable number of references and studies on this topic, the authors draw on only 5 references published within the last three years out of a total of 21. At least 40% of the works cited should be included in the review of current manuscripts. This review of the state of the art affects the discussion of the research.
On the method, the authors describe the sample, but should provide more information on the recruitment of the study participants and the selection criteria of the participants. On the other hand, the authors should present the aim of the study before starting the method section or at the beginning of this section.
Similarly, the authors present an elementary quantitative study based on the analysis of socio-demographic data of the participants. However, this study is not justified to a large extent as an added value to the methodology, since these findings serve perfectly well to describe characteristics of the sample. Therefore, these characteristics could be included in a "participants" section within the "method".
As for the qualitative results, a quantitative analysis of an analysis of themes associated with four broad categories, which in turn are distributed into 11 sub-themes, is offered. Very interesting testimonies from the participants are given below. However, the authors need to provide a more in-depth explanation of the qualitative methodology used. They should also have addressed essential aspects in this type of study, such as the validity of the study and the reliability of the analyses carried out.
Author Response
This paper is of interest in exploring the risk factors and barriers to providing substitute family care for children whose parents have mental illness. To conduct this study, the authors use a descriptive quantitative and qualitative research design with an interesting sample of 94 participants (2.54% of the eligible participants), all of whom are substitute family caregivers in India. The authors collect caregivers' perceptions of the economic, family and social risks and barriers associated with caregiving. The majority of the substitute caregivers were women from low-income households. They suggest the identification of 11 specific sub-themes associated with the risks and barriers of substitute care. These sub-themes are grouped into four broad themes: economic, family, school-related risks, and cultural and service access barriers. It concludes with the need for a paradigm shift of the treatment approach from the patient to the whole household to benefit children and patients.
Author responses: Thank you for agreeing to review our manuscript. Your valuable suggestions are greatly appreciated, and we have modified the manuscript based on your comments
On the other hand, the literature review is interesting, and both primary and secondary scientific sources are collected. However, although the paper succeeds in addressing an acceptable number of references and studies on this topic, the authors draw on only 5 references published within the last three years out of a total of 21. At least 40% of the works cited should be included in the review of current manuscripts. This review of the state of the art affects the discussion of the research.
Author responses: The introduction section has been updated with newer references.
On the method, the authors describe the sample but should provide more information on the recruitment of the study participants and the selection criteria of the participants. On the other hand, the authors should present the aim of the study before starting the method section or at the beginning of this section.
Author responses: A small paragraph on participant recruitment and selection criteria are added after the study design.
The aim of the study is presented just before the methods section as a short paragraph, as suggested.
Similarly, the authors present an elementary quantitative study based on the analysis of the socio-demographic data of the participants. However, this study is not justified to a large extent as an added value to the methodology, since these findings serve perfectly well to describe the characteristics of the sample. Therefore, these characteristics could be included in a "participants" section within the "method".
Author responses: Thank you for your suggestion. The quantitative table is moved to the methods section under “sample”.
As for the qualitative results, a quantitative analysis of an analysis of themes associated with four broad categories, which in turn are distributed into 11 sub-themes, is offered. Very interesting testimonies from the participants are given below. However, the authors need to provide a more in-depth explanation of the qualitative methodology used. They should also have addressed essential aspects of this type of study, such as the validity of the study and the reliability of the analyses carried out.
Author responses: A detailed explanation on data analysis and the methods used is added to the manuscript.
Reviewer 3 Report
1. As mentioned in the article, Caregivers include mother, father, grandparent and other relatives, what are the differences in risks and obstacles faced by different substitute carer for children? Are there any corresponding solutions in the policy recommendations?
2. The description of respondents in 2.1. Design is not clear, and it is not clear whether the respondents in this study are mental patients or surrogate caregivers for their children. Are the selected respondents representative? 2.3 What is the Data mining tool? A brief introduction to the data mining tool is recommended, not just the demographic information filtered
3. What is the difference between Carer and caregiver? Which is the substitute caregiver for the child and which is the caregiver for the mentally ill? Or are they the same group? It would have been better to suggest a common noun. In addition, the caregiver (or carer) related variables in Table 1 are clearer in the same section
4. Risks and obstacles are divided into four themes: economy, family, school risks, culture and service barriers. However, the theme of Impact of economic drift is included in Table 2. What is the logical relationship between the Impact of economic drift and Economic risk factors? In addition, the study is Substitute family care, what is the significance of analyzing school risks and cultural barriers? What is the main contribution of this article?
Author Response
Thank you for agreeing to review our manuscript. Our responses to the valuable suggestions are given below.
As mentioned in the article, Caregivers include mother, father, grandparent and other relatives, what are the differences in risks and obstacles faced by different substitute carer for children? Are there any corresponding solutions in the policy recommendations?
Author response: we have included it in the discussion section.
- The description of respondents in 2.1. Design is not clear, and it is not clear whether the respondents in this study are mental patients or surrogate caregivers for their children. Are the selected respondents representative?
Author response: Our respondents were family members who were providing primary care to children who had a parent with mental illness. A subsection on participant recruitment is added.
2.3 What is the Data mining tool? A brief introduction to the data mining tool is recommended, not just the demographic information filtered
Author response: The data mining tool essentially includes the information available from hospital records, and it is largely centred around socio-demographics. The content of the tool is described.
What is the difference between Carer and a caregiver? Which is the substitute caregiver for the child, and which is the caregiver for the mentally ill? Or are they the same group? It would have been better to suggest a common noun. In addition, the caregiver (or carer) related variables in Table 1 are clearer in the same section
Author responses: Both words are used interchangeably. To ensure uniformity, we have updated the entire manuscript to use caregivers (instead of carers).
We are only focusing on the substitute caregiver for the child than the caregiver for the person with mental illness. In some cases, both the primary caregivers were the same even.
- 4. Risks and obstacles are divided into four themes: economy, family, school risks, culture and service barriers. However, the theme of Impact of economic drift is included in Table 2. What is the logical relationship between the Impact of economic drift and Economic risk factors?
Author responses: We have modified the manuscript for better clarity. However, we would like to present our rationale for separating them into two. The first part (Economic risk factors) dealt with the various aspects of economic risks, and the second part (Economic drift) was the impact of economic risks on the family. However. The single subhead – The economic risk factors cover both dimensions. Therefore, we deleted the second major head and included the findings under the global theme “Economic risk factors”.
In addition, the study is Substitute family care, what is the significance of analyzing school risks and cultural barriers? What is the main contribution of this article?
Author responses: We included the school barriers to demonstrate that if the school is supportive, it would be protective against family stress, especially the worry regarding the child’s future. As academic achievements are prerequisites for better jobs, higher income, better living conditions and, thereby, economic well-being. Therefore, a supportive school system would be a great resource for the substitute caregiver.
Regarding the cultural obstacles, we changed the title to be more accurate and consistent with the main theme—shifting the treatment's emphasis from the patient to the family. Collective societies, like India, have strong family and community ties; if supportive, the members feel social cohesion, which fosters trust, a sense of safety, and involvement in the community. Social inclusion is essential for lowering stress levels, facilitating better access to resources for addressing material deprivations, and ensuring a sense of support.
Round 2
Reviewer 2 Report
I endorse the manuscript.
Reviewer 3 Report
Accept in present form